# The Mining for Flowering-Related Genes Based on De Novo Transcriptome Sequencing in the Endangered Plant *Phoebe chekiangensis*

**DOI:** 10.3390/ijms26031000

**Published:** 2025-01-24

**Authors:** Qinglin Sun, Yan Liu, Mingyang Ni, Yandong Song, Qi Yang, Junhong Zhang, Yuting Zhang, Zaikang Tong

**Affiliations:** 1State Key Laboratory of Subtropical Silviculture, School of Forestry & Biotechnology, Zhejiang A&F Univesity, Hangzhou 311300, China; sunqinglin555@gmail.com (Q.S.); liuy12345671@gmail.com (Y.L.); 15158063607@163.com (M.N.); qiyang@zafu.edu.cn (Q.Y.); zktong@zafu.edu.cn (Z.T.); 2Lishui Institute of Agriculture and Forestry Sciences, Lishui 323000, China; songyandong21@163.com

**Keywords:** *Phoebe chekiangensis*, transcriptome, de novo assembly, flowering

## Abstract

*Phoebe chekiangensis* is an indigenous, endangered, and valuable timber and garden tree species in China, which is notable for having a short juvenile phase (early flowering), unique among the *Phoebe* genus. However, the molecular mechanisms regulating the flowering of *P. chekiangensis* remain unexplored, primarily due to the lack of transcriptomic or genomic data. In the present study, transcriptome sequencing yielded 53 million RNA reads, resulting in 111,250 unigenes after de novo assembly. Of these, 47,525 unigenes (42.72%) were successfully annotated in the non-redundant (Nr) database. Furthermore, 15,605 unigenes were assigned to Clusters of Orthologous Groups (KOGs), and 36,370 unigenes were classified into Gene Ontology (GO) categories. A total of 16,135 unigenes were mapped to the Kyoto Encyclopedia of Genes and Genomes (KEGG) database, involving 298 pathways. Based on the expression levels, Gibberellin signaling pathway-related genes were the most predominant expression levels. Hormonal analysis showed that gibberellin (GA) levels varied across tissues and flowering stages, as GA20 levels in leaves were low during full bloom, while GA1 and GA5 levels peaked in flowers. Furthermore, several key genes involved in gibberellin biosynthesis, including *CPS*, *GID1*, *GA20ox*, *GA3ox*, and *GA2ox*, exhibited stage-specific expression patterns. Certain genes were highly expressed during the initial phases of flowering, while others, like *GA3ox* and *GA2ox*, reached peak expression at full bloom. These findings provide valuable insights into the molecular mechanisms underlying flowering in *P. chekiangensis*, laying the foundation for future breeding efforts. This transcriptome dataset will serve as an important public resource for molecular research on this species, facilitating the discovery of functional genes related to its growth, development, and flowering regulation.

## 1. Introduction

*Phoebe chekiangensis*, a member of the Lauraceae family, is native to subtropical China. The *Phoebe* genus is renowned for its durable wood, unique fragrance, and attractive golden-tinted surface, making it highly valued in high-grade furniture manufacturing, shipbuilding, temple decoration, and its use in historical sites such as the Forbidden City [1]. This wood, often referred to by foreigners as “Chinese Magical Wood”, symbolizes a deep cultural tradition. *P. chekiangensis* also has ornamental value due to its tall, straight trunk and evergreen, majestic canopy. However, its natural distribution is limited to Zhejiang, Jiangxi, and Fujian provinces. Overexploitation and poor natural regeneration have led to a decline in its population, prompting its inclusion as a category II protected plant in China’s “List of Wild Plants Under State Protection” [2,3].

Research on *P. chekiangensis* has primarily focused on chromosome karyotype analysis [4], container seedling cultivation, seedling growth, photosynthetic characteristics [5], and genetic diversity using ISSR and SSR markers [6]. However, there is limited research on the molecular biology regulating growth and development, especially flowering regulation. While most Phoebe species have long vegetative phases of over 12 years, *P. chekiangensis* and *P. sheareri* begin flowering at around four years of age. The timing of flowering is critical for plant reproductive success [7], and identifying flowering-related genes is crucial for understanding the transition from vegetative to reproductive growth in *P. chekiangensis*.

Flowering in plants marks a vital developmental switch when the shoot apical meristem starts producing flowers instead of leaves [8]. In *Arabidopsis thaliana*, around 200 genes are involved in controlling flowering time, with many genes interacting across a network of six major pathways: the photoperiod and vernalization pathways, which respond to seasonal changes in day length and temperature; the ambient temperature pathway; and the age, autonomous, and gibberellin (GA) pathways, which function more independently of environmental cues [9]. These pathways converge on a small number of “floral integrator genes”, such as *FLOWERING LOCUS T* (FT), *SUPPRESSOR OF OVEREXPRESSION OF CONSTANS 1* (SOC1), and *LEAFY* (LFY), which rapidly promote flowering. Beyond model plants, significant advances in flowering regulation have been made in other species, including *Artemisia annua* [10], *Glycine max* [11], *Solanum melongena* [12], and *Pyrus communis* [13].

Plant hormones, including auxin, cytokinins, Abscisic Acid (ABA), and gibberellins (e.g., GA1, GA5, GA20), play pivotal roles in regulating plant growth, development, and responses to environmental stress [14]. These hormones influence key physiological processes, such as cell division, fruit maturation, and adaptation to environmental challenges like water scarcity [15,16,17]. Gibberellins (GAs), a diverse family of plant hormones, are particularly crucial for regulating various aspects of growth, including leaf expansion, stem elongation, flowering, and fruit development [18]. The GA pathway is one of the six primary pathways influencing flowering and has been extensively studied, particularly in model plants [19]. Key enzymes such as KAO, KS, CPS, and KO are involved in GA biosynthesis, while GA2ox, GA20ox, and GA3ox regulate GA metabolism, maintaining appropriate hormone levels [20]. The GA signaling pathway also relies on the soluble GA receptor GID1, which plays an essential role in mediating GA responses. GA biosynthesis occurs in three stages across the plastid, endoplasmic reticulum, and cytoplasm, culminating in the production of bioactive GAs like GA1 and GA4, which are prevalent in higher plants [21,22]. Genes encoding enzymes such as GA2ox, GA3ox, and GA20ox, which form part of the oxygenase family, have been identified in various species, including Arabidopsis, rice, and tomato [23,24,25].

In this study, we measured the hormone levels in the leaves and flowers of *P. chekiangensis* at different stages and identified the crucial role of gibberellins in the flowering process. RNA-seq and de novo assembly yielded 111,250 unigenes, with a substantial portion annotated in databases like Nr, KOG, GO, and KEGG, covering 298 pathways. Based on these promising results, we identified the genes involved in gibberellin biosynthesis and performed quantitative analysis on their expression, further confirming the role of gibberellins in the flowering regulation of *P. chekiangensis*.

## 2. Results

### 2.1. Determination of Hormone Content During P. chekiangensis Flowering Stages

Using HPLC-MS/MS, we quantified the concentrations of several endogenous hormones (Figure 1). IAA (Indole-3-Acetic Acid), a type of auxin, exhibits an increasing trend in concentration as leaves develop, though differences between consecutive stages are not statistically significant. During the late stage of floral organ development, its concentration peaks at approximately 24 ng/g, significantly higher than at other stages. Except for the F3 stage, there is no significant difference in IAA levels between floral organs and leaves. This pattern may be attributed to changes in auxin biosynthesis, transport, or degradation during specific developmental stages.

Zeatin, a cytokinin, is most abundant during the L1 leaf stage and decreases as leaves progress through the L2, L3, and L4 stages. In floral organs, the expression pattern of zeatin mirrors that in leaves, with concentrations declining as the organs develop. Similarly, Trans-Zeatin-Riboside (TZR) exhibits the highest concentration in L1, with significantly lower levels in L2. In floral organs, the concentration of TZR is significantly higher in F1 compared to F2 and F3. This decline in later stages could result from reduced demand for cell division as the organs mature.

In leaves, Isopentenyladenine (IP), a soluble cytokinin, peaks at the L2 stage, with significantly lower concentrations in L1, L3, and L4. In floral organs, IP levels are significantly lower during F2 compared to F1 and F3. However, its concentrations in leaves and floral organs are similar overall, showing no significant differences.

N6-Isopentenyladenosine (IPA), a bound cytokinin, is significantly lower during the L3 leaf stage compared to other leaf stages. In floral organs, IPA levels decline progressively, reaching their lowest in F3. During the F1 stage, IPA concentrations are comparable to those in leaves, while, at F2, levels are lower than in L1, L2, and L4 but higher than in L3.

ABA concentrations in leaves show no significant differences across L1–L3 stages but decrease significantly at L4. In floral organs, ABA levels decline progressively with development. This reduction may result from a diminished demand for ABA’s role in suppressing cell division as organs mature and enter later developmental stages.

GA1, a type of gibberellin, shows no significant variation in concentration across leaf stages but is significantly lower in leaves compared to floral organs. In floral organs, GA1 levels do not vary significantly between stages. GA5 is most abundant during the F2 floral stage, while its concentration in L4 leaves is the lowest. In leaves, GA5 levels show no significant differences across stages. However, in floral organs, F2 levels are significantly higher than F1 and F3, though there is no significant difference between F1 and F3. Finally, GA20 is most concentrated at the L1 stage, with no significant differences observed between L2, L3, and L4. In floral organs, GA20 levels at L1 are significantly higher than F1 and F2 but show no significant difference from F3.

### 2.2. Sequencing Analysis and De Novo Assembly

Due to the lack of a reference genome for *P. chekiangensis*, de novo transcriptome sequencing was conducted. This sequencing generated 61,231,742 raw reads, which, after preprocessing, yielded 52,527,954 clean reads with an average length of 94.31 bp. The GC content of these clean reads exhibited a normal distribution with a mean of 45.91% (Figure 2a), and the length distribution is shown in Figure 2b.

A total of 111,250 unigenes were identified, with 48,587 (43.67%) aligned to the TREMBL database and 47,525 (42.72%) to the Nr database. Among these, 87.91% of the unigenes aligned with species in the top 10 list, with the highest similarity to *Vitis vinifera* (62%), followed by *Populus trichocarpa* and *Ricinus communis* (Appendix A).

Among the annotated unigenes, 27 exhibited RPKM values exceeding 500, representing the most abundant transcripts. These included genes predicted to encode metallothionein-like protein type 2, glycine-rich RNA-binding protein GRP1A, lipid transfer protein, ubiquitin C, and glutathione S-transferase zeta class (Appendix A).

Gene Ontology (GO) analysis assigned GO terms to 36,370 of the 111,250 unigenes, which were categorized into three primary groups: biological process, cellular component, and molecular function (Figure 2c). These categories were further divided into 60 subcategories (Appendix A). The most enriched terms in the biological process category included “metabolic process” (GO:0008152, 22.33%) and “cellular process” (GO:0009987), which encompassed 21,235 unigenes. In the cellular component category, “cell” (GO:0005623) and “cell part” (GO:0044464) were most enriched. For molecular function, “binding” (GO:0005488) and “catalytic activity” (GO:0003824) represented 22.21% and 17.84% of the unigenes, respectively.

To gain deeper insight into biological pathways, we aligned all assembled unigenes to the KEGG database. This analysis mapped 16,135 unigenes to various KEGG pathways, with 948 enzyme commission (EC) numbers assigned, spanning a total of 298 pathways (Appendix A). The most represented pathways included ribosome (ko03010, 531 unigenes), protein processing in the endoplasmic reticulum (ko04141, 496 unigenes), plant hormone signal transduction (ko04075, 443 unigenes), plant–pathogen interaction (ko04626, 426 unigenes), and spliceosome (ko03040, 410 unigenes). Additionally, notable pathways such as starch and sucrose metabolism (ko00500, 396 unigenes), purine metabolism (ko00230, 363 unigenes), RNA transport (ko03013, 361 unigenes), and oxidative phosphorylation (ko00190, 343 unigenes) were identified (Figure 2d).

In a separate approach to predict gene functions and evaluate the completeness of the transcriptome library, we analyzed all unigenes against the KOG database (Appendix A). In total, 15,605 unigenes were annotated into 25 KOG categories (Figure 2e). The largest category was *Signal transduction mechanisms*, encompassing 3680 unigenes (23.58%). In contrast, only five unigenes were classified into the *Cell motility* category.

### 2.3. Identification of Flowering-Related Genes in P. chekiangensis

Through transcriptomic analysis, we identified several putative homologs of key enzymes that play critical roles in gibberellin (GA) biosynthesis and signal transduction. These genes are associated with essential steps in the GA metabolic pathway and include those encoding GA20ox, GA2ox, and GA3ox, which are pivotal for the synthesis and catabolism of active GAs. Additionally, we detected components integral to GA signaling, such as the F-box protein GID2, the gibberellin receptor GID1, and DELLA proteins, which mediate downstream responses to GA signaling (Figure 3).

The expression patterns of these GA-related genes, measured by their RPKM values, exhibited a broad range of activity, reflecting variable roles and regulatory controls within the pathway. Some genes demonstrated strong transcriptional activity, while others were expressed at relatively lower levels, suggesting functional specialization or tissue-specific regulation.

In contrast to the findings on GA-related genes, our analysis did not uncover homologous genes involved in the autonomous and vernalization pathways that are well-characterized in *Arabidopsis*. This absence in the *P. chekiangensis* transcriptome suggests potential divergence in flowering regulatory mechanisms between these species.

### 2.4. The Expression Patterns of Flowering-Related Genes

The expression analysis revealed that the early-stage GA biosynthesis genes exhibited widespread expression across all analyzed organs (Figure 4a). Specifically, the genes KAO (Ketenol-acid oxidase) and KO (Kaurene oxidase) in leaves displayed similar expression patterns, peaking at the L2 stage. In contrast, the expression of KS (Kaurene synthase) in leaves reached its highest levels during the L2 stage and then declined from the F1 to F3 stages. For CPS (Copalyl diphosphate synthase), there were no significant changes from L1 to L3, but its expression peaked in the F2 stage in flowers.

The expression patterns of GA2ox (Gibberellin 2-oxidase) and GA3ox (Gibberellin 3-oxidase) in leaves followed a consistent trend from L1 to L3, with an initial increase followed by a decrease. However, their expression in flowers showed a contrasting pattern. GA2ox in flowers initially increased and then decreased, while GA3ox exhibited the opposite trend. GA20ox (Gibberellin 20-oxidase) exhibited a similar expression pattern to GA2ox and GA3ox, showing a consistent downward trend throughout the L period. In flowers, GA20ox displayed the same expression pattern as GA2ox.

The expression pattern of GID (Gibberellin Insensitive Della) in the L1 to L3 stages was similar, with no significant differences observed. However, in the flower organs, GID levels were highest at the F2 stage, and significant differences in expression were found between the F1, F2, and F3 stages.

The correlation analysis revealed that GA3ox, KAO, and KS exhibited a strong positive correlation with GA20 biosynthesis, but a negative correlation with the synthesis of GA1 and GA5. In contrast, CPS showed a positive correlation with GA1 and GA5 biosynthesis, while it was negatively correlated with GA20 synthesis. GID displayed a similar pattern to CPS, although its correlation with the biosynthesis of GA1 and GA5 was weaker than that of CPS.

## 3. Discussion

### 3.1. Plant Hormones and Phase Transitions in Growth

Plant hormones play essential roles in regulating growth and development, especially by driving the phase transitions that guide plants through critical life stages [15,17]. Auxins, including IAA, IP, and IPA, are crucial regulators in the transition from vegetative to reproductive growth [26]. By influencing cell division, elongation, and differentiation, auxins maintain apical dominance and promote lateral root development, thereby supporting the formation of a robust root system to ensure resource availability during early growth [27,28]. For instance, the increase in IAA triggers the initiation of the first flowering period in *C. subhirtella* ‘Autumnalis’ before its mandatory hibernation [29]. Auxin accumulation also upregulates the expression of APETALA1 (CmAP1), subsequently activating inflorescence primordium development in the axillary buds of *Castanea mollissima* [30]. In *P. chekiangensis*, IAA, IP, and IPA exhibit increasing concentrations in flower tissues, underscoring their vital roles in flower development and maturation. Conversely, in flowers and leaves, these compounds are present at lower concentrations, indicating their functions are more critical in flower development.

Cytokinins, such as zeatin and trans-zeatin riboside (TZR), primarily stimulate cell division and growth, impacting floral organ development and fruit quantity [31,32]. For example, transgenic expression of AtIPT4 elevated cytokinin levels in *Jatropha curcas*, resulting in an increased flower number and improved female-to-male ratio, despite flower infertility [33]. Additionally, Gn1a encodes OsCKX2, an enzyme responsible for degrading cytokinin in rice, and its reduced expression leads to increased cytokinin levels, enhancing reproductive organ development and grain yield [32]. In *P. chekiangensis*, zeatin and TZR exhibit similar expression trends, peaking during the L1 leaf stage, indicating their critical roles in cell division and growth in young leaves. However, their levels decrease in mature leaves and flower samples, suggesting a minimal impact on flower development and the transitions between developmental phases.

ABA regulates multiple facets of plant physiology, such as inducing seed dormancy and facilitating adaptive responses to both abiotic and biotic stresses [34,35]. Most studies on ABA focus on its roles in plant physiology, including promoting seed dormancy, facilitating adaptive responses to stresses, and regulating growth and development mechanisms [34,36]. The decline in ABA levels alleviates the inhibition of NnSnRK1, triggering flower bud abortion and providing new insights into the regulation of flowering timing in lotus [37]. In *P. chekiangensis*, ABA levels trend higher in early leaves and flowers, while levels decrease in later leaves and flowers. This pattern suggests that ABA may be crucial in the initial growth and development stages, influencing stress responses and the transition to later developmental phases.

Gibberellins, particularly GA1, have emerged as key regulators in these processes, exerting a notably stronger influence on developmental transitions than other hormones such as IAA and cytokinins, which play comparatively minor roles [38,39]. Our analysis of *P. chekiangensis* highlights the impact of these hormones during pivotal stages like flower maturation. The pronounced peak of GA1 observed in the flower samples suggests that it not only promotes physical growth but is also vital for reproductive timing. Known for driving cell elongation and division, gibberellins like GA1 are crucial for flower induction and fruit ripening, enabling the cellular changes necessary for full fruit maturity [18]. Additionally, elevated GA1 levels in *P. chekiangensis* underscore its importance in ensuring robust flower development. Beyond these direct effects, gibberellins interact with other hormones, such as auxins and cytokinins, regulating flowering and fruit set [39,40]. This suggests a coordinated hormonal network that collectively supports the transition from vegetative to reproductive growth. The interplay between gibberellins, auxins, and cytokinins likely modulates various developmental stages, enabling the plant to respond flexibly to internal and external cues [15,16]. Based on the aforementioned research, gibberellins may play a significant role in the transition of *P. chekiangensis* from vegetative to reproductive growth. Insights into the synthesis mechanisms and regulatory patterns of gibberellins could provide valuable information for breeding efforts in *P. chekiangensis*.

### 3.2. Gibberellin Biosynthesis Pathway, Key GA-Related Gene Identification, and Their Interrelationships

The gibberellin (GA) biosynthetic pathway is crucial for plant growth and development, beginning with the production of geranylgeranyl pyrophosphate (GGPP) from isopentenyl pyrophosphate (IPP) through the action of isopentenyl pyrophosphate synthase [41]. This GGPP is then converted into gibberellin acid by gibberellin acid synthase, marking the initiation of gibberellin synthesis. Subsequently, this gibberellin acid undergoes hydroxylation to generate GA20 and GA19 [42,43]. Enzymes such as GA20 oxidase and GA3 oxidase further convert these intermediates into the biologically active gibberellins GA1 and GA3, which are essential for key processes such as seed germination, floral bud differentiation, fruit development, and stem elongation [44]. The enzymes KO (ent-kaurene oxidase) and KAO (ent-kaurenoic acid oxidase) play pivotal roles in modifying gibberellin structures, thus influencing their biological activity [45,46]. The primary gibberellin receptor, GID1, regulates these functions by activating signaling pathways that modulate the expression of growth-related genes [40]. Additionally, C3 hydroxylase (GA2ox) converts active gibberellins into inactive forms, helping maintain hormonal balance within the plant [47]. The intricate synthesis and signaling of gibberellins are affected by various environmental factors, allowing plants to adapt to changing conditions and optimize growth.

Through de novo assembly of the transcriptome from *P. chekiangensis,* we identified key genes involved in gibberellin (GA) biosynthesis and signaling pathways, which were further validated through qPCR experiments. The expression levels of CPS, GID, GA2ox, and KO were significantly higher at the F2 stage compared to other genes, and the GA5 content also peaked at this stage. Correlation analysis revealed that the synthesis of GA5 was most strongly correlated with CPS, suggesting that CPS plays a crucial role in GA5 biosynthesis in *P. chekiangensis.*

## 4. Materials and Methods

### 4.1. Plant Material

The experimental materials of *P. chekiangensis* were sourced from Qingyuan Forest Farm, Lishui, China, with trees of the same age selected for this study(Figure 5). Sampling was performed every half month after the initial collection, covering four stages of leaf development and three floral organs with distinct differences. The leaves were labeled from L1 to L4, and the floral organs were labeled from F1 to F3. Each sample included three biological replicates. All samples were used for RNA-seq and hormone content analysis. Three distinct stages of leaves (L1, L2, and L3) and floral organs (F1, F2, and F3) with clear differences were used for qPCR experiments. For all the above experiments, three technical replicates were performed for each biological replicate.

### 4.2. Analysis of Endogenous Hormone Content

Endogenous hormone content was analyzed using an AB Qtrap 6500 mass spectrometer (AB Sciex, Framingham, MA, USA) operating in triple quadrupole-ion mode. Quantitative hormone analysis was conducted using the ESI-HPLC-MS/MS method. Sample separation was achieved with an Agilent 1290 high-performance liquid chromatograph (Agilent Technologies, Santa Clara, CA, USA), utilizing electrospray ionization (ESI) as the ion source. Detection was performed in multiple reaction monitoring (MRM) mode. Hormone concentrations were analyzed using one-way analysis of variance (ANOVA) to evaluate the effects of sample groups. Post hoc pairwise differences between groups were identified using Tukey’s Honest Significant Difference (HSD) test.

### 4.3. RNA Extraction and qPCR

Total RNA was extracted using Trizol reagent (Promega, Madison, WI, USA), followed by treatment with RNase-free DNase to remove genomic DNA, verified using 1% agarose gel electrophoresis. cDNA was synthesized from RNA using the PrimeScript™ RT reagent kit (TaKaRa Biotechnology, Dalian, China) according to the manufacturer’s instructions in a final volume of 20 μL. Target genes were quantified using qPCR (CFX96, BioRad, Hercules, CA, USA) with SYBR Premix Ex Taq (TaKaRa Biotechnology, Dalian, China) in a total reaction volume of 10 μL, following the manufacturer’s protocol. The primers used for qPCR are listed in Appendix A. The thermal cycling conditions included an initial step at 95 °C for 30 s, followed by 40 cycles of 95 °C for 5 s and 60 °C for 20 s. A dissociation curve was generated from 60 to 95 °C, and the melting curve exhibited a single peak. All qPCR reactions were performed in technical triplicate. The qPCR data were analyzed to assess significant differences in gene expression levels across the samples, with EF1α serving as the reference gene. A one-way analysis of variance (ANOVA) was conducted, using expression levels as the dependent variable and sample groups as the independent factor. Subsequently, Tukey’s Honest Significant Difference (HSD) test was applied as a post hoc analysis to identify pairwise differences between the groups. A *p*-value < 0.01 was considered indicative of significant differences between the samples.

### 4.4. cDNA Library Construction and High-Throughput RNA Sequencing

RNA samples were sent to LC Science (Hangzhou, China) for cDNA library construction and RNA sequencing. Briefly, mRNA was enriched from 10 μg of total RNA using oligodT25 magnetic beads, then fragmented into short sequences. First-strand cDNA synthesis was performed using random hexamer primers, followed by second-strand synthesis with a dUTP mix. Double-stranded cDNA was purified using magnetic beads and subjected to end repair and 3′ adenylation. Sequencing adaptors were ligated to the adenylated fragments, followed by size selection and enrichment via PCR amplification. The quality and quantity of the sample library were assessed using an Agilent 2100 Bioanalyzer (Agilent Technologies, Santa Clara, CA, USA). The paired-end libraries, with an average insert size of approximately 200 bp, were sequenced on an Illumina Hiseq2000 (Illumina, San Diego, CA, USA), following the manufacturer’s recommended protocols, with a sequencing depth of 30×.

### 4.5. Data Filtering, De Novo Assembly, and Annotation

Raw reads were preprocessed using Trimmomatic with default parameters to remove low-quality reads and Illumina adapters, resulting in clean reads. The reads were then de novo assembled using Trinity with default parameters, and the assembly quality was evaluated through length distribution, N50, and average length analyses. The best candidate coding sequences (CDSs) for each contig were identified, resulting in a set of Unigenes. These Unigenes were annotated via BLASTx alignment (E-value < 1 × 10^−5^) against public protein databases, including Nr, Swiss-Prot, and TREMBL. Gene Ontology (GO) terms were assigned using Blast2GO Basic 6.0, categorizing them into molecular function, cellular component, and biological process. Metabolic pathways were identified using the Kyoto Encyclopedia of Genes and Genomes (KEGG) database. Transcript abundance (Unigene) was calculated in reads per kilobase per million mapped reads (RPKM) using the Bowtie 0.12.8 software. The E-value threshold used ensures the reliability of functional annotations.

### 4.6. Identification of GA Pathway Genes and Correlation Analysis with GA Content

Flowering-related genes were identified based on standard gene names and synonyms in the Unigene sequence annotations. The results were further validated through BLAST searches against the NCBI database (E-value < 1 × 10^−5^). Genes with results identical to the keyword searches were identified in *P. chekiangensis*, and their roles in flowering pathways were subsequently mapped. To evaluate the relationship between gene expression levels and gibberellin (GA) content, Spearman rank correlation analysis was performed. Spearman rank correlation coefficients were calculated based on gene expression data and GA content data to assess the correlation between gene expression and GA content. Statistical significance was determined using a *p*-value (values less than 0.05 were considered significant). All data analyses were conducted using R version 4.3.1, with the cor.test () function used to calculate Spearman.

## 5. Conclusions

This study highlights the pivotal roles of plant hormones, particularly gibberellins (GA), in regulating key developmental transitions in *P. chekiangensis*, revealing that GA biosynthetic genes such as KO, KAO, GA2ox, and GA3ox exhibit distinct expression patterns during flowering stages, underscoring their critical contributions to reproductive development, while auxins and cytokinins support vegetative growth and initial reproductive transitions, offering insights into the hormonal networks that govern growth, stress responses, and developmental phase shifts, thereby providing a foundation for breeding strategies to enhance fruit quality and yield in *P. chekiangensis*.

Looking ahead, future research should focus on elucidating the functional roles of the identified GA-related genes and their interactions with other hormonal pathways, including those involving photoperiod and vernalization. Investigating these pathways could provide valuable insights into the adaptive strategies employed by *P. chekiangensis*, especially in response to varying environmental conditions. Furthermore, understanding the regulatory mechanisms of gibberellins in this species could inform breeding and conservation efforts, ultimately enhancing our broader comprehension of plant developmental biology and its evolutionary implications. By exploring these avenues, we can deepen our knowledge of how *P. chekiangensis* and other plants optimize their growth and reproductive success in diverse ecological contexts.

## Figures and Tables

**Figure 1 ijms-26-01000-f001:**
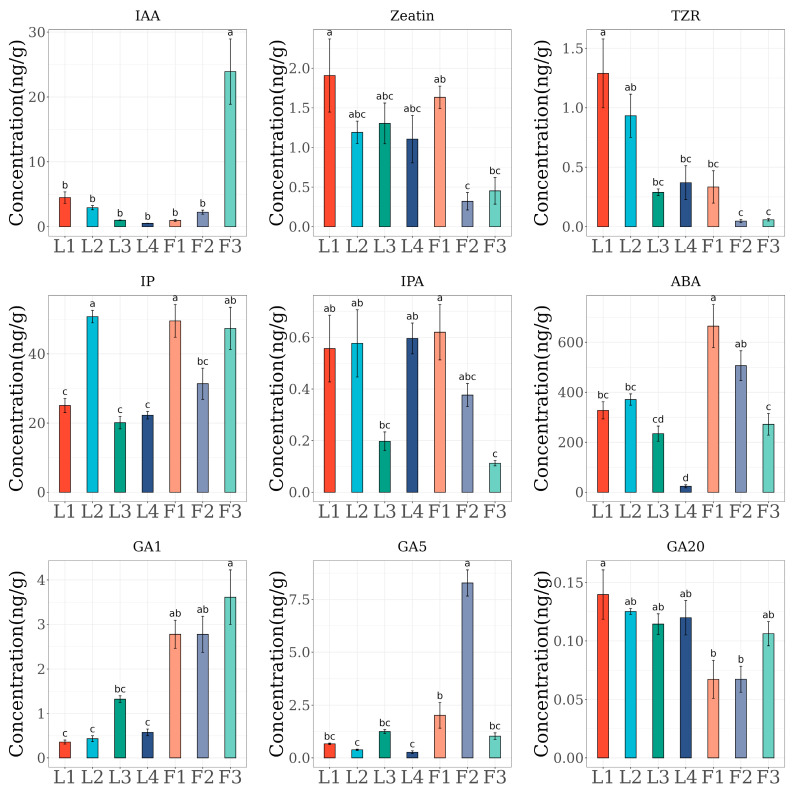
The hormone content in different samples. *p* values were calculated using Tukey’s HSD test, and different letters indicate a *p* value of < 0.01.

**Figure 2 ijms-26-01000-f002:**
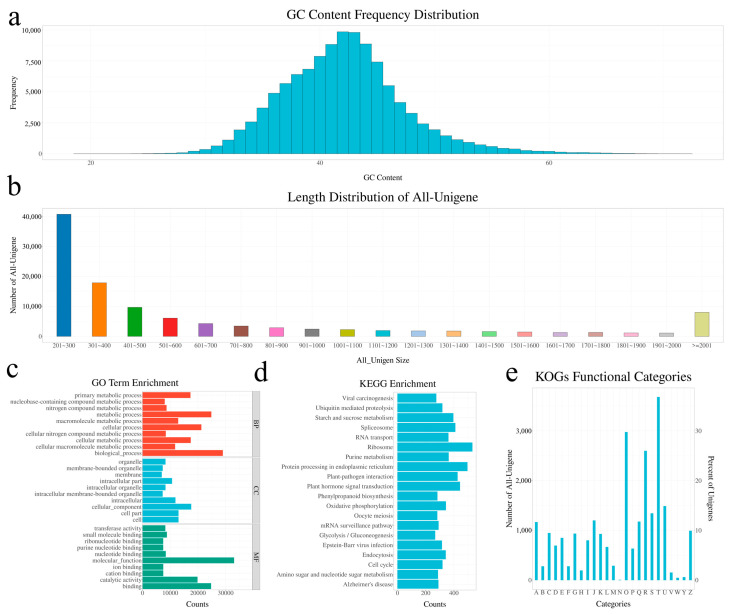
Transcriptome sequencing and annotation results of *P. chekiangensis.* (**a**) GC content and distribution: The GC content distribution of the assembled unigenes, indicating the overall sequence composition. The majority of unigenes show a balanced GC content, ranging from 40 to 50%. (**b**) Length distribution of all unigenes: the unigene length distribution, with a significant number of unigenes in the 300–1000 bp range, reflecting the diversity of gene sizes in the transcriptome. (**c**) GO enrichment analysis: Gene Ontology (GO) term categorization, highlighting enriched functions in biological processes, molecular functions, and cellular components. (**d**) KEGG annotation analysis: KEGG pathway mapping showing the involvement of unigenes in various metabolic and signaling pathways. (**e**) KOG annotation results: KOG functional classification revealing key roles in signal transduction, protein modification, and general cellular functions.

**Figure 3 ijms-26-01000-f003:**
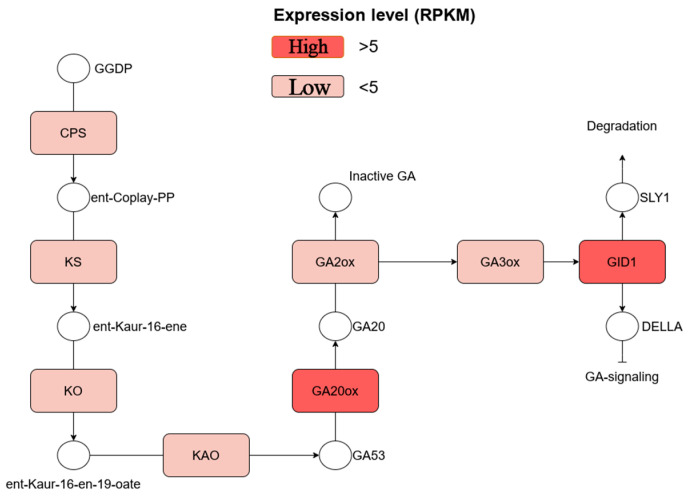
Genes involved in the GA biosynthesis pathway in *P. chekiangensis* and their RPKM values.

**Figure 4 ijms-26-01000-f004:**
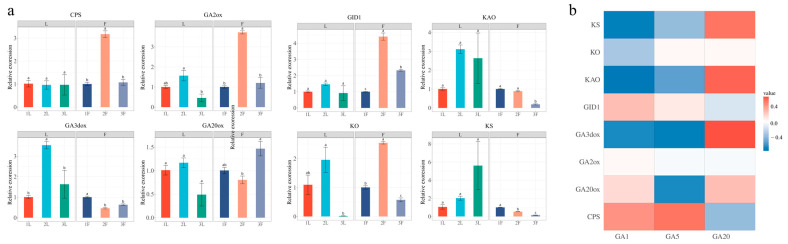
qPCR of genes in the gibberellin biosynthesis pathway and their correlation with gibberellin. (**a**) Relative expression levels of genes in the gibberellin biosynthesis pathway. *p* values were calculated using Tukey’s HSD test, and different letters indicate a *p* value < 0.01. (**b**) Correlation analysis between genes involved in the GA biosynthesis pathway and GA content.

**Figure 5 ijms-26-01000-f005:**
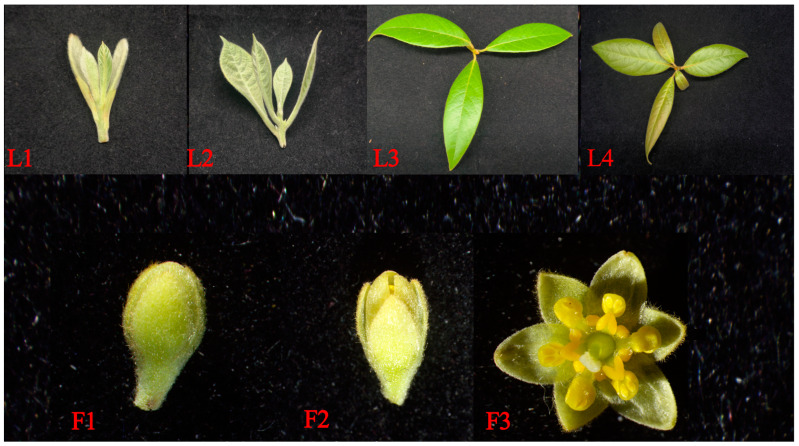
Materials were sourced from Qingyuan Forest Farm. The first row represents leaves at different developmental stages, labeled L1, L2, L3, and L4, from left to right. The second row represents floral organs at different developmental stages, labeled F1, F2, and F3, from left to right.

## Data Availability

Data are contained within the article and Appendix A.

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
