# Peer review of "The Mining for Flowering-Related Genes Based on De Novo Transcriptome Sequencing in the Endangered Plant Phoebe chekiangensis"

_ijms, 2025, doi:10.3390/ijms26031000_

Round 1

Reviewer 1 Report

Comments and Suggestions for Authors

This manuscript presents an interesting study in gene expression analysis of the flower develop process in Phoebe chekiangensis.

However in stead of the wok developed manuscript showed important deficiencies affecting the quality of the obtained results..

Results of the Figure 1 does not correspond ith the experimental design of the Material and Methods section. Authors must clarify the ,meaning of the assayed samples.

Results of the Figure 4 does not correspond ith the experimental design of the Material and Methods section. Authors must clarify the ,meaning of the assayed samples.

Quality of Figure 2 is very poor. The legend also must be stribglky revised to be self-redeable.

In the description of the results in section 2.2 authors must describe the repositoty and the bioproject qith the reads and the obtained results.

Correwlation between phenology, hormones and gene expression is very confuse and nedd a more intensive analysis to be discussed.

Experimental design is poorly described in the Section 4.1 and Figure 5. Author must clarify the assayed tissues and the technical and biological replications evaluated.

Additionally, a detailed description of the time, the period and the climatic conditions of the assya must be included. At this moment this experimental design represeting the flowering development is very poor.

Analysis of the hormone contents is ver week. The method and the assayed hormones must be clarified. Numbre of replications is missing.

qPCR analysis must also be beeter described. refernce genes and statistical analysis must be incorporated. In this analysis and in the RNA-Seq study authros must clarify biological and technical replications.

A better description of the RNA-Seq analysi must be introduced.

Author Response

1. Summary

Thank you for handling our review process and for the reviewers' comments on our manuscript entitled " The mining for flowering-related genes based on de novo transcriptome sequencing in the endangered plant Phoebe chekiangensis" (ijms-3370882). These comments were valuable and very helpful. We have carefully studied these comments and have made changes that we hope will meet with your approval. Based on the instructions you provided in your letter, we have uploaded the files of the revised manuscript. We have made revisions to the manuscript and the changes in the text are highlighted in yellow to improve the presentation at all levels of the manuscript. We have endeavoured to respond carefully to the reviewers' comments.

2.Point-by-point response to Comments and Suggestions for Authors

Comments 1: Results of the Figure 1 does not correspond with the experimental design of the Material and Methods section. Authors must clarify the, meaning of the assayed samples.

Response 1: Thank you for your reminder. Wecollected leaves samples at four different development stages and three floral organs at different stages flowering . The samples were used for analyzing plant hormone content and related-genes expression levels. Relevant details have been added to the plant materials section Line 310-323

Comments 2: Results of the Figure 4 does not correspond with the experimental design of the Material and Methods section. Authors must clarify the ,meaning of the assayed samples.

Response 2: Thank you for your suggestion. We used three stages of leaf samples and floral organ samples with notable differences in GA content for qPCR analysis. Relevant details have been added to the plant materials section:“The qPCR data were analyzed to assess significant differences in gene expression lev-els across the samples, with EF1α serving as the reference gene. A one-way analysis of variance (ANOVA) was conducted, using expression levels as the dependent variable and sample groups as the independent factor. Subsequently, Tukey's Honest Significant Difference (HSD) test was applied as a post hoc analysis to identify pairwise dif-ferences between the groups. A p-value < 0.01 was considered indicative of significant differences between the samples.”

Comments 3: Quality of Figure 2 is very poor. The legend also must be stribglky revised to be self-readable.

Response 3: Thank you for your suggestion. We have redrawn Figure 2, enlarged the image, removed unnecessary legends, and added more detailed annotations to the image content in lines 171-180: ' Transcriptome sequencing and annotation results of P. chekiangensis.(a) GC content and distribution: The GC content distribution of the assembled unigenes, indicating the overall sequence composition. The majority of unigenes show a balanced GC content, ranging from 40-50%.(b) Length distribution of all unigenes: The unigene length distribution, with a significant number of unigenes in the 300-1000 bp range, reflecting the diversity of gene sizes in the transcriptome.(c) GO enrichment analysis: Gene Ontology (GO) term categorization, highlighting enriched functions in biological processes, molecular functions, and cellular components.(d) KEGG annotation analysis: KEGG pathway mapping showing the involvement of unigenes in various metabolic and signaling pathways.(e) KOG annotation results: KOG functional classification revealing key roles in signal transduction, protein modification, and general cellular functions.'

Comments 4: In the description of the results in section 2.2 authors must describe the repository and the bioproject with the reads and the obtained results.

Response 4: Thank you for pointing this out. The data used here were  from the article "Development of EST-SSR markers and analysis of genetic diversity in natural populations of endemic and endangered plant Phoebe chekiangensis." Relevant information can be found in this source.

Comments 5: Correlation between phenology, hormones and gene expression is very confuse and need a more intensive analysis to be discussed.

Response 5: We sincerely appreciate you pointing this out. We have added content in Section 2.4 to describe the relationship between plant hormones and gene expression in Figure 4b. The analysis results demonstrate the correlation between different genes and the three types of GA content, clarifying the association between genes and plant hormones.

Comments 6: Experimental design is poorly described in the Section 4.1 and Figure 5. Author must clarify the assayed tissues and the technical and biological replications evaluated.

Response 6: Thank you for your valuable suggestions. We collected samples from different plant tissues, including leaves, roots, and stems, for RNA-seq, aiming to obtain as complete mRNA data as possible. Since our focus is on plant hormones that affect flowering, we selected leaves and floral organs for plant hormone content analysis and qPCR analysis. Each experiment included three biological and technical replicates, and the RNA-seq sequencing depth was 30x. Relevant descriptions have been added to the appropriate sections Line 320-323

Comments 7: Additionally, a detailed description of the time, the period and the climatic conditions of the assya must be included. At this moment this experimental design represeting the flowering development is very poor.

Response 7: Thank you for your reminder. We collected leaf and floral organ samples every two weeks between March and April, resulting in four stages of leaf samples and three stages of floral organ samples. All samples were used for plant hormone content analysis. Since the GA pathway is a key flowering-related pathway, we selected samples with significant differences in GA levels—three stages of leaves (L1-L3 in Figure 5) and three stages of floral organs (F1-F3 in Figure 5)—for qPCR analysis, with the aim of identifying genes involved in GA biosynthesis. This information has been added to the Materials section.

Comments 8: Analysis of the hormone contents is very week. The method and the assayed hormones must be clarified. Number of replications is missing.

Response 8: Thank you for your suggestions. Endogenous hormone content was analyzed using an AB Qtrap 6500 mass spectrometer in triple quadrupole-ion mode, with separation performed on an Agilent 1290 HPLC. Detection was carried out in MRM mode, and quantification was achieved using the ESI-HPLC-MS/MS method. Hormone concentrations were analyzed by one-way ANOVA, with pairwise differences identified using Tukey's HSD test. Each sample included three biological and technical replicates. We have revised the hormone content analysis section and provided specific details regarding the samples used for each experiment in the plant materials section.

Comments 8: qPCR analysis must also be better described. reference genes and statistical analysis must be incorporated. In this analysis and in the RNA-Seq study authros must clarify biological and technical replications.

Response 8: We sincerely appreciate you pointing this out. qPCR data were analyzed to assess gene expression differences, using EF1α as the reference gene. One-way ANOVA was performed with expression levels as the dependent variable and sample groups as the independent factor. Tukey's HSD test was then used for pairwise comparisons, and a p-value < 0.01 was considered significant. For RNA-seq, we have three biological replicates with a sequencing depth of 30x. We have added the corresponding descriptions in the relevant sections line 344~350

Comments 9: A better description of the RNA-Seq analysi must be introduced.

Response 9: Thank you for your valuable suggestions. We have updated the relevant description of the RNA-seq process in M&M sections line 364-375. “Raw reads were preprocessed using Trimmomatic with default parameters to re-move low-quality reads and Illumina adapters, resulting in clean reads. The reads were then de novo assembled using Trinity with default parameters, and the assembly quality was evaluated through length distribution, N50, and average length analyses. The best candidate coding sequences (CDS) for each contig were identified, resulting in a set of Unigenes. These Unigenes were annotated via BLASTx alignment (E-value <1E-5) against public protein databases, including Nr, Swiss-Prot, and TREMBL. Gene Ontology (GO) terms were assigned using Blast2GO, categorizing them into molecular function, cellular component, and biological process. Metabolic pathways were identi-fied using the Kyoto Encyclopedia of Genes and Genomes (KEGG) database. Transcript abundance (Unigene) was calculated in reads per kilobase per million mapped reads (RPKM) using Bowtie 0.12.8 software. The E-value threshold used ensures the reliabil-ity of functional annotations.”

Reviewer 2 Report

Comments and Suggestions for Authors

The presented work highlights the relevance of the study and characterization of endangered plants to establish their biotechnological importance and ecological relevance at biochemical and molecular levels. The flowering process is vital to understanding the reproductive mechanisms at the molecular level and designing effective strategies for plant conservation in the environment.

There are some minor corrections to be addressed in the manuscript:

In Figure 1, the legend for "samples" should be removed, and the figure footnote for the statistical analysis (ANOVA and post-hoc test) applied must be included, as well as the significance difference code.

In line 168, change "(3.31%" to "(23.58%)". 

In Figure 3, the RPKM values indicated are very low; I suggest to change for "low" (<5) and "high" (>5).

In line 205, change "trajectory" to "trend".

In Figure 4, remove the legend for "Samples". Add the statistical test (ANOVA and post-hoc test) applied and the significance difference code in the Figure footnote.

In line 353, after "low-quality reads", add the Phred value used as the threshold.

In line 367, the E-value threshold used is adequate.

In line 368, turn "P. chekiangensis" into italics.

In the Materials and Methods section, 4.2 and 4.3 subsections, please add statistical analysis was applied to determine significant differences.

Author Response

1. Summary

Thank you for handling our review process and for the reviewers' comments on our manuscript entitled " The mining for flowering-related genes based on de novo transcriptome sequencing in the endangered plant Phoebe chekiangensis" (ijms-3370882). These comments were valuable and very helpful. We have carefully studied these comments and have made changes that we hope will meet with your approval. Based on the instructions you provided in your letter, we have uploaded the files of the revised manuscript. We have made minor revisions to the manuscript and the changes in the text are highlighted in yellow to improve the presentation at all levels of the manuscript. We have endeavoured to respond carefully to the reviewers' comments.

Comments 1: In Figure 1, the legend for "samples" should be removed, and the figure footnote for the statistical analysis (ANOVA and post-hoc test) applied must be included, as well as the significance difference code.

Response 1: Thank you for pointing this out. We first performed an analysis of variance (ANOVA), followed by a Tukey's HSD test to calculate the P-value. A P-value less than 0.01 was considered statistically significant, and different letters were used to annotate the plot. This information has been added to the footnote of the figure line 130-131:’ P values were calculated using Tukey's HSD test, and different letters indicate a p value < 0.01. ’.

Comments 2: In line 168, change "(3.31%" to "(23.58%)".

Response 2: We sincerely appreciate you pointing this out. We have corrected this error in the text line 168.

Comments 3: In Figure 3, the RPKM values indicated are very low; I suggest to change for "low" (<5) and "high" (>5).

Response 3:Thank you for your suggestion. We have updated the figure and added text descriptions to correspond to the colors.

Comments 4: In line 205, change "trajectory" to "trend".

Response 4: We appreciate your careful review and have addressed this error in the text line 209.

Comments 5: In Figure 4, remove the legend for "Samples". Add the statistical test (ANOVA and post- hoc test) applied and the significance difference code in the Figure footnote.

Response 5: Thank you for pointing this out and we have acted accordingly. The legend has been removed from Figure 1, and an explanation of the differential analysis method (as described in Response 1), along with the meaning of the significance markers, has been added to the footnote line 226-229.

Comments 6: In line 353, after "low-quality reads", add the Phred value used as the threshold.

Response 6: Thank you for your reminder. As this step was carried out using the software's default parameters, we have revised the phrasing of this section accordingly: ‘Raw reads were preprocessed using Trimmomatic with default parameters to re-move low-quality reads and Illumina adapters, resulting in clean reads. The reads were then de novo assembled using Trinity with default parameters, and the assembly quality was evaluated through length distribution, N50, and average length analyses’.

Comments 7: In line 367, the E-value threshold used is adequate.

Response 7: In Section 4.6 (the part on line 367),, we performed a preliminary identification of relevant genes based on the annotation results of the transcriptome assembly. The identified genes were then submitted to the NCBI BLAST function for further validation and were confirmed as target genes. An E-value < 1e-5 is the default parameter for NCBI BLAST. We are not entirely sure about your question and would appreciate more specific guidance from you.

Comments 8: In line 368, turn "P. chekiangensis" into italics.

Response 8: We appreciate your reminder and have resolved this issue.

Comments 9: In the Materials and Methods section, 4.2 and 4.3 subsections, please add statistical analysis was applied to determine significant differences.

Response 9: Thank you for your suggestion. All differential analyses were first conducted using analysis of variance (ANOVA), followed by a Tukey's HSD test to calculate the P-value. A P-value less than 0.01 was considered to indicate a significant difference. A more detailed description has been added to Sections 4.2 and 4.3 in line 336-338 and line 350-356 : ‘Hormone concentrations were analyzed using one-way analysis of variance (ANOVA) to evaluate the effects of sample groups. Post hoc pairwise differences between groups were identified using Tukey's Honest Significant Difference (HSD) test’ and ‘The qPCR data were analyzed to assess significant differences in gene expression levels across the samples, with EF1α serving as the reference gene. A one-way analysis of variance (ANOVA) was conducted, using expression levels as the dependent variable and sample groups as the independent factor. Subsequently, Tukey's Honest Significant Difference (HSD) test was applied as a post hoc analysis to identify pairwise differences between the groups. A p-value < 0.01 was considered indicative of significant differences between the samples.’

Round 2

Reviewer 1 Report

Comments and Suggestions for Authors

Authors have reviewed correctly the manuscript